# Resilient Behaviors in Music Students: Relationship with Perfectionism and Self-Efficacy

**DOI:** 10.3390/bs13090722

**Published:** 2023-08-30

**Authors:** Félix Arbinaga

**Affiliations:** Department of Clinical and Experimental Psychology, University of Huelva, 21007 Huelva, Spain; felix.arbinaga@dpsi.uhu.es

**Keywords:** effectiveness, musician, music conservatory, perfection, resilience

## Abstract

Self-efficacy and perfectionism play an important role in high-performance activities. This cross-sectional study analyzes the relationship between these constructs and resilience in a sample of 145 music students (57.9% female) with a mean age of 27.77 years. Perfectionism was assessed using the Multidimensional Inventory of Perfectionism in Sport; resilience, using the Resilience Scale; and self-efficacy, using the General Self-Efficacy Scale. Females, compared to males, are more perfectionist, both on the adaptive (Cohen’s *d* = 0.41) and maladaptive scales (Cohen’s *d* = 0.70). However, no gender differences were found in self-efficacy or resilience scores. Music students categorized as highly resilient obtained significantly higher self-efficacy scores (Cohen’s *d* = 1.30). However, no differences were found between high- and low-resilience students in perfectionism scores, the total scale scores, or its adaptive or functional factor (striving for perfection). Differences were found for the maladaptive factor, negative reactions to imperfection, where low-resilience students scored higher on negative reactions to imperfection (Cohen’s *d* = 0.49). Self-efficacy shows significant predictive power for resilience (*β* = 0.525, *p* < 0.001). Although functional perfectionism did not significantly predict resilience, a marginal negative relationship was found between dysfunctional perfectionism and resilience (*β* = −0.156, *p* = 0.063). The results are discussed concerning their implications for music pedagogy and teacher intervention.

## 1. Introduction

Music—whether performed by professionals or students—is an activity associated with various problems that, if not properly managed, can hinder the healthy development of professional and artistic careers [1].

Aside from facing the academic demands of the discipline, music students undergo long training sessions. These training routines often involve repetitive movements that are practiced in the pursuit of perfection. This regimen is physically stressful and cognitively taxing for students, which increases their vulnerability to physical fatigue, pain, psychological distress, injury, and dropout [2].

All these demands mean musicians must develop coping skills and strategies to manage the adversities they will face during their training and later professional life. In this regard, resilient behavior is relevant [3,4]. The resilience construct refers to the cognitive, social, motor, and emotional behaviors through which challenges and new circumstances are faced [5]. It facilitates functional adaptation to adverse environments with minimal consequences [6], so low scores are associated with problems in the context of performance [7,8].

Other competencies and skills recognized as relevant to high performance include self-efficacy [9,10] and perfectionism [11,12,13]. The concept of self-efficacy was developed based on two types of expectations: efficacy and outcome [14,15]. General self-efficacy refers to people’s stable beliefs in their ability to adequately handle various daily life stressors [16]. Abundant literature supports the idea that self-efficacy—derived from experiences during the practice of a motor task—predicts outcomes on subsequent learning tests [9,17,18,19] or performance output [10,20,21].

Research has shown a significant relationship between self-efficacy and actual performance on music assessment tests [20]. Furthermore, a clear superiority of self-efficacy over other variables as a predictor of performance has been observed in these musical performance situations [20,22].

Research exploring the relationship between self-efficacy and resilience reveals that a strong sense of self-efficacy is important for maintaining high levels of resilience [23]. The importance of self-efficacy and its relationship with resilience has been supported in adolescents and minors [24,25], where greater self-efficacy facilitates the ability to cope with unfamiliar situations and adapt to new circumstances. Furthermore, such relationships facilitate adult leadership qualities [26], while resilience is strengthened through enhancing and developing protective factors such as self-efficacy [27,28]. In summary, it appears that high self-efficacy helps to increase ego resilience [29]. Furthermore, resilience has also been shown to be a good predictor of self-efficacy [30]. Thus, research indicates that high resilience scores are necessary to develop high levels of self-efficacy in, for example, dance students [31].

Perfectionism has been understood from a multidimensional perspective. The characteristics that define perfectionism are high personal demands and negative self-evaluation [12,32,33,34]. The former component—adaptive perfectionism [34]—has been related to positive characteristics, processes, and outcomes in subjective well-being, good psychological adjustment, challenge appraisals, and active coping [35]. In contrast, the negative self-evaluation component has been identified as essentially dysfunctional or maladaptive. For example, Khawaja and Armstrong [34] associate maladaptive perfectionism with a preoccupation with mistakes, doubts about actions, and critical parental expectations.

The music education context presents very specific characteristics for studying perfectionism since this is often linked to the performer’s perception of perfection rather than a truly perfect performance [36]. The student’s interpretation of success is determined by the tastes of the audience, which consists of teachers, peers, family members, and others [37]. It has been found that in students, this perfectionist tendency can also develop under the influence of “other-oriented perfectionism”, as observed in dance teachers, who impose the demand for perfection on their students, both technically and artistically [11].

Research on perfectionism in music students has primarily focused on various facets of the construct, such as perfectionism and music performance anxiety, job stress, trait anxiety, academic procrastination, motivation, effort and achievement orientations, and family factors in the development of perfectionism, coping skills, and social phobias [13,38,39,40,41,42,43]. 

However, there is a scarcity of research examining the relationship between perfectionism and resilient behaviors. The few published studies indicate that individuals with high scores on socially prescribed perfectionism tend not to use resilient skills [44,45,46], while self-oriented and socially prescribed perfectionists are prone to catastrophizing [47] and dependency [48], characteristics that are not highly resilient. Other studies reported that only socially prescribed perfectionism was negatively associated with resilience, whereas self-oriented and other-oriented perfectionism was not significantly related to resilience scores [46]. Finally, some evidence suggests that maladaptive perfectionism is negatively associated with resilience [49].

While research linking these constructs to resilient behaviors has been relatively scarce, investigations exploring the interrelationship among the three constructs have been practically non-existent. In this regard, the only published study to jointly analyze the three constructs was conducted on a sample of soccer referees, with linear regression analyses indicating how self-efficacy and adaptive perfectionism significantly predict resilience scores, with the weight of self-efficacy being greater than that of perfectionism [50].

In this context, the present study seeks to investigate the relationship between the constructs of self-efficacy and perfectionism and the resilient behaviors shown by music students. The importance of these constructs lies in their role in execution and performance and the fact that these are skills susceptible to modification through intervention. Music students, like professionals, are subjected to strong pressures from the environment (academic and/or musical) that make them a high-performance group with significant performance demands. They are a group where the analyzed constructs are relevant to cope adequately with daily demands. In this sense, they present characteristics of perfectionism (criteria of perfectionism, socio-educational environment on which they build perfectionism, etc.) that make them a peculiar performance group for which it is important to analyze the constructs proposed. As a first hypothesis, highly resilient music students are expected to obtain higher self-efficacy scores than their low-resilience counterparts. Our second hypothesis predicts that music students with high resilience scores will also show higher scores on adaptive or functional perfectionism. Adaptive perfectionism is considered to allow progress in high performance. In this sense, it is predictable that it is necessary for resilient behaviors to be effective and efficient, since adversities are always present in performance activities. The third hypothesis states that music students with low resilience will obtain higher scores on dysfunctional perfectionism than those with high resilience scores. This prediction arises when, after not achieving the objectives, attention and resources are focused on critically analyzing errors, without positive contributions. This will require greater difficulty and activation to reorganize the behavior of the musician. Finally, the fourth hypothesis predicts that self-efficacy scores correlate positively with functional perfectionism and negatively with dysfunctional or maladaptive perfectionism.

## 2. Materials and Methods

### 2.1. Participants

This was a cross-sectional study. The sample consisted of 145 music students (57.9% female) with a mean age of 27.77 years (SD = 14.95). In 57.9% of the cases, the students specialized in string instruments, whereas 33.8% specialized in wind instruments, and 8.3% specialized in percussion instruments. The inclusion criteria were the following: (1) to be a music student under the guidance of a teacher and not self-taught, (2) to have undergone training in a music academy or a conservatory, (3) to have been in training for a minimum of three years and always with a teacher, (4) to be over 18 years of age, and (5) to give informed consent.

### 2.2. Instruments

Through ad hoc interviews, information was collected on demographic variables (year of birth and gender), musical activity (years practicing music, rehearsal days per week with a teacher, weekly rehearsal hours with a private teacher/academy/conservatory, weekly rehearsal hours without a teacher, and the main type of instrument usually played). The instruments considered were wind, string, percussion, and electric instruments.

To measure perfectionism, we used the short version of the Multidimensional Inventory of Perfectionism in Sport (MIPS) [51] in its Spanish version adapted by Pineda-Espejel et al. [52]. This instrument comprises ten items that begin with the following phrase, which is adapted to the musical context: “During rehearsal or performance in a show…” Five items assess Factor 1 (F1), “striving for perfection” (e.g., “I have the desire to do everything perfectly”), and the remaining five items assess Factor 2 (F2) “negative reactions to imperfection” (e.g., “I feel completely furious if I make mistakes”). The Likert-type response scale ranges from never (1) to always (6). The measure of total perfectionism was calculated through combining the scores of the striving for perfection and negative reactions to imperfection items. The scale has shown good internal consistency in the present study, assessed using Cronbach’s alpha, both for the total score (*α* = 0.897) and the dimensions of perfectionism: striving for perfection (*α* = 0.896) and negative reactions (*α* = 0.890).

Resilient behavior was assessed using the “Resilience Scale” (RS) [3,4] in its Spanish adaptation by Ruiz-Barquín et al. [53]. The RS contains two factor scores and a total score. Factor 1 (F1) refers to “personal competence” and comprises items such as self-confidence, decision-making, and perseverance. Factor 2 (F2) refers to the “acceptance of self and life” and measures adaptability, balance, flexibility, and a stable life perspective that coincides with an acceptance of life and a feeling of peace despite adversity. With these two factors—F1 and F2—five areas of resilience are represented (personal satisfaction, feeling good alone, self-confidence, stability, and perseverance). The instrument consists of 25 items, where respondents assign a score to each item from 1 (disagree) to 7 (totally agree). Thus, the total factor score (TF) ranges between 25 and 175 points, with high scores indicative of good resilience [4]. According to these authors, the scores can be categorized to indicate low resilience (<147 RS points) and high resilience (≥147 RS points). In this work, the scale has shown good internal consistency, as assessed using Cronbach’s alpha, both for the total score (TF) (*α* = 0.877) and for the F1 “personal competence” (*α* = 0.857) and F2 “acceptance of self and life” (*α* = 0.715) factors.

Self-efficacy was assessed using the “General Self-Efficacy Scale” (GSES) [16] in the Spanish version by Sanjuán-Suarez et al. [54]. This scale evaluates perceptions of personal competence to handle demanding situations and obtain the expected outcomes. Higher scores indicate greater self-efficacy. In the present study, the scale obtained a Cronbach’s alpha of 0.833. 

### 2.3. Procedure

The data were collected in paper format through visiting three music conservatories and five private academies in different cities. At the same time, a mail was sent to the directors of conservatories and music academies in other cities, requesting their collaboration and distribution of the link to the questionnaire among the students. At the beginning of the tests, the objectives of the research, the legal terms, and the informed consent were described. As part of this study, rigorous data validation processes were implemented to ensure the quality and reliability of the information used in the analysis. Various stages of data verification and cleaning were conducted to identify potential errors, outliers, and missing data. Imputation techniques were applied to address missing values systematically, thereby ensuring that the analysis results were representative and robust. Additionally, exploratory data analysis was performed to detect any unusual patterns or inconsistencies. These validation procedures guarantee data integrity and support the soundness of the conclusions drawn from this study.

### 2.4. Data Analysis

An a priori power analysis was conducted using G*Power-3 [55] to determine the minimum sample size required to test the study hypothesis. The results indicated that the sample size required to achieve 95% power to detect a mean effect, with a significance criterion of *α* = 0.05, was N = 147 for Student’s *t*-test for independent groups. Therefore, the obtained sample size of N = 145 is adequate to test the study hypothesis.

The following was carried out: Descriptive analyses (frequencies, percentages, means, and standard deviation) were conducted to characterize the main research variables. Normality tests (kurtosis and skewness) of the variables were performed according to the proposal made by Munro [56]. The reliability of the tests was calculated using Cronbach’s alpha (*α*). The comparison of quantitative variables was carried out using Student’s *t*-test for independent groups. The effect size was estimated using Cohen’s *d* (*d* < 0.2—small effect size; *d* = 0.2 to 0.8—medium effect size and *d* > 0.8—large effect size). In the case of categorical variables, the chi-squared test (*χ*^2^) was used. For categorical variables, Cramer’s V was used to estimate the effect size (<0.2—small effect size; between 0.2 and 0.6—moderate effect size and >0.6—large effect size). 

Associations between the variables were analyzed using Pearson correlations, and stepwise linear regression analysis was employed to determine the predictors of resilience. All analyses were conducted using the SPSS statistical package (IBM ver. 20.0, SPSS Inc., Armonk, NY, USA).

## 3. Results

As seen in Table 1, there were no age differences in the participants according to gender. The participants indicated that they have been practicing with the musical instrument for more than 11 years, with no gender differences. In addition, the participants report practicing an average of two days a week, playing the instrument for around three and a half hours a week with the teacher and nine hours independently. No differences were observed in these variables as a function of gender. However, a marginal gender difference was observed regarding the type of instrument they specialize in, with more women choosing stringed instruments as their specialty.

An analysis of the normality of the variables and a consideration of kurtosis and skewness values revealed that the data show a normal distribution, respecting the criterion interval [−1.96; 1.96] proposed by Munro [56]. In this regard, it was found that, of the ten values presented, only one did not fit the interval required to confirm the normality of the distribution. However, this mismatch will not affect the statistical tests considered in the analyses. Thus, the three variables analyzed can be assumed to follow a normal distribution: self-efficacy (kurtosis: 0.254/skewness: −0.256); perfectionism (kurtosis: −0.632/skewness: −0.028); F1—high personal demands (kurtosis: −0.714/skewness: −0.386); F2—negative self-evaluation (kurtosis: −0.887/skewness: 0.024); and resilience (kurtosis: 3.157/skewness: −1.091).

It was found that female music students obtained higher scores on the perfectionism scale than males (see Table 2), with a medium effect size (Cohen’s *d* = 0.66).

Females are shown to be more perfectionist overall and, according to the scores on each of the subscales, striving for perfection, adaptive perfectionism (Cohen’s *d* = 0.41), and negative reactions or dysfunctional perfectionism (Cohen’s *d* = 0.70). However, no gender differences were found in self-efficacy or resilience scores.

Music students categorized as highly resilient obtained significantly higher self-efficacy scores (see Table 3) with a large effect size (Cohen’s *d* = 1.30). However, no differences were found between high- and low-resilience students in perfectionism scores, the total scale scores, or its adaptive or functional factor (F1, striving for perfection). Differences were found for the maladaptive factor F2, negative reactions to imperfection, where low-resilience students scored higher on negative reactions to imperfection, with a medium effect size (Cohen’s *d* = 0.49).

Table 4 shows the bivariate Pearson correlations between the three constructs. 

Resilience scores for the scale’s total score or each factor show significant and positive correlations with self-efficacy scores. Similarly, Factor 1 of the resilience scale (personal competence) shows positive and significant correlations with Factor 1 of the perfectionism scale, which indicates adaptive or functional perfectionism. However, Factor 2 of the resilience scale (acceptance of self and life) shows significant negative correlations with total perfectionism scores and dysfunctional or negative reactions to imperfection. Finally, it should be noted that self-efficacy shows a positive correlation with adaptive perfectionism (*p* = 0.048).

The results of predictive models of resilience based on self-efficacy, functional perfectionism, and dysfunctional perfectionism as predictor variables can be observed in Table 5. It appears that self-efficacy explains 27.5% of the variance in resilience scores (*p* < 0.001), with a predictive power of *β* = 0.525 (*p* < 0.001). When the striving for perfection or functional perfectionism variable is added to the model, there is a slight increase in its explanatory power, reaching 27.8% (*p* < 0.001); however, the self-efficacy construct loses some of its predictive power *β* = 0.516 (*p* < 0.001), and Factor 1 of perfectionism does not contribute significantly to its predictive capacity. Finally, when Factor 2 of perfectionism, or maladaptive perfectionism, is introduced into the model, its explanatory power increases, reaching 29.5% but maintaining a marginal significance level (*p* = 0.063). However, the introduction of self-efficacy reduces the model’s predictive capacity (*β* = 0.486), although it remains significant (*p* < 0.001). Factor 1 of perfectionism does not contribute significantly to the model’s predictive capacity, while dysfunctional perfectionism (Factor 2) shows marginal predictive power but with a negative sign (*β* = −0.156).

## 4. Discussion

This research has sought to contribute to and expand the existing knowledge on how the constructs of self-efficacy and perfectionism are related to the resilient behaviors shown by music students. Four hypotheses have been proposed, the first being that highly resilient music students would show higher self-efficacy scores than those with low resilience. Our second hypothesis predicted that music students with high resilience scores would also obtain higher adaptive or functional perfectionism scores. The third hypothesis stated that low-resilience students would obtain higher scores in dysfunctional perfectionism than their highly resilient counterparts. Finally, the fourth hypothesis predicted that scores on the self-efficacy test would correlate positively with functional perfectionism and negatively with dysfunctional or maladaptive perfectionism scores.

Our data support the first hypothesis since we observed that music students with high resilience scores showed higher self-efficacy scores. In this regard, it was also found that self-efficacy maintains a high predictive capacity for resilient behaviors. However, it is difficult to establish the directionality of the causal relationship between these two constructs. 

Notably, our findings support those of other studies in the literature showing that self-efficacy is important for maintaining the high efficacy of resilient behaviors [23] in adolescents, minors, and adults [24,25,26]. Furthermore, it has already been mentioned that self-efficacy facilitates coping with novel, unfamiliar situations and obtaining effective adaptation outcomes, while resilient skills are strengthened by enhancing factors such as self-efficacy [27,28], since this helps to increase the resilience of the ego [29]. Similarly, and reversing the directionality of the relationship, resilience is a good predictor of self-efficacy; high scores in resilience facilitate the development of high levels of self-efficacy in students of performance activities such as dance [30,31].

Our second hypothesis—that music students with high resilience scores would score higher on adaptive or functional perfectionism—was not borne out by our results since no differences were observed between the high- and low-resilience groups. Indeed, the possible existence of “true” perfectionism traits in some performance activities, such as dance [21], has already been questioned. However, it is understood that there are several common elements of perfectionism, particularly those associated with its maladaptive characteristics.

This absence of significant group differences might be explained if we consider that professionals and students are transmitted the message that perfection is achieved through striving for continuous improvements in performance. Therefore, regardless of resilient behaviors, all students would display this effort toward reaching goals centered around achieving sustained and incremental improvements in performance. Some authors have reported differences in positive or functional perfectionism across different high-performance activities [57]. For example, in a sample of dance students, differences were observed in their reaction to errors rather than in the expectations of achieving optimal performance [21]. However, contrary to the results observed in the present study, where functional perfectionism does not predict or explain resilient behavior in music students, a study with soccer referees [50] found that adaptive perfectionism significantly predicted resilience scores, although to a lesser extent than self-efficacy.

Our third hypothesis stated that music students with low resilience would obtain higher scores in dysfunctional perfectionism than those with high resilience scores. Our data supported this prediction. Moreover—and as mentioned above—these findings support the line of argument developed in the literature where it was considered that differences could be observed in the reaction to errors or so-called dysfunctional perfectionism [21]. Thus, it has been suggested that people with high scores in socially determined perfectionism tend to use non-resilient skills [44,45,46]. For example, self-directed and socially prescribed perfectionists have shown greater dependence and catastrophizing [47,48], characteristics not associated with resilience. Furthermore, socially prescribed perfectionism was negatively associated with resilience, whereas self-oriented and other-oriented perfectionism was not significantly related to resilience scores [46]. These findings align with what was found in our sample of music students, suggesting that high-resilience behaviors predict lower scores in the negative reaction to imperfection.

Finally, our fourth hypothesis predicted that self-efficacy scores would correlate positively with functional perfectionism and negatively with dysfunctional or maladaptive perfectionism. This hypothesis has not been completely supported since adaptive perfectionism significantly correlated with self-efficacy but not with dysfunctional perfectionism. However, these data align with those of a similar study conducted on young students in a school setting [58], where adaptive perfectionism traits correlated positively with self-efficacy in study and commitment.

Although non-significant, dysfunctional perfectionism showed a negative correlation with resilience, which might indicate a trend in the relationship between these constructs, as observed in some studies showing that maladaptive perfectionism was negatively associated with resilience and self-efficacy [49]. Similarly, the dysfunctional component of perfectionism is often associated with perceived discrepancies between expected and actual performance, increasing concerns about mistakes, uncertainty about one’s actions, and parental criticism, all of which predict worse future performance [34].

The research may present certain limitations that should be analyzed for future work. A possible limitation is the type of study carried out; being correlational, it does not allow causal relationships between variables to be established. On the other hand, the sample size should be increased in future studies. In this sense, it would be necessary to work with more homogeneous samples of music students and/or professionals. This would allow us to determine the role played by each of the constructs analyzed, depending on the type of student or professional. Similarly, it is necessary to monitor, with a more rigorous approach, the level of training, performances, demands, and future expectations of the students. It should also be noted that it would be interesting for future work to analyze the relationships that may exist between self-concept, self-efficacy, and perfectionism.

For future research, it is necessary to establish experimental designs to define the causal relationships between the type of perfectionism (adaptive/maladaptive) and resilient behaviors, as some researchers have argued that higher performance may be a factor contributing to the development of perfectionism in children and adolescents [59]. In addition, it is necessary to understand how maladaptive perfectionism interferes with or hinders performance (in the short and long term) and the adjustment to situations with adverse outcomes. In this regard, it is important to note the number of research studies that support the benefits of perfectionism, provided that perfectionistic efforts are not accompanied by high levels of perfectionistic concerns. It has been suggested that perfectionistic concerns are unhealthy and maladaptive and can pose a serious risk to people’s well-being and mental health.

On the other hand, it is necessary to control for factors such as the influence of the teacher (other-oriented perfectionism) [11] and the type of performance of the participants or the type of instruments used, since differences have been shown depending on the instrument considered [60]. Future research could examine whether the interaction between types of perfectionism and coping strategies differs as a result of a musician’s experience [43].

## 5. Conclusions

Music is a high-performance activity associated with various problems that can hinder a musician’s professional and artistic career. These problems make it necessary to develop effective coping skills for managing challenging situations adaptively. To this end, conservatories and academies are spaces that can play a key role in promoting such preventive strategies.

Constructs such as self-efficacy, perfectionism, and resilience are important for high performance, not only due to their impact on execution and performance but also because these skills are amenable to modification through appropriate intervention strategies. It should be noted that if a teacher can help students to shift their emphasis from perfectionism toward the pursuit of excellence, they will be aligning their students’ goals with the results of current research on perfectionism, motivation, and goal-setting. To this end, academic institutions should seek to prioritize the promotion of self-oriented or functional perfectionism while minimizing the influence of other-oriented, socially prescribed, or dysfunctional perfectionism.

## Figures and Tables

**Table 1 behavsci-13-00722-t001:** Social and music-related characteristics of the participants according to gender.

	Total	Men	Women	*t* _(_ * _df_ * _=143)_	*p*
	145	61 (42.4)	84 (57.9)		
Age	27.77 (14.95)	28.34 (15.69)	27.36 (14.47)	0.391	0.696
Years of practice (meses)	135.88 (71.87)	146.08 (85.74)	128.48 (59.29)	1.462	0.146
Days/week rehearses with teacher	2.26 (1.47)	2.21 (1.49)	2.29 (1.46)	0.293	0.770
Rehearsal time/week (min)					
With teacher	228.10 (245.19)	190.08 (170.74)	255.71 (285.31)	1.725	0.087
Without teacher	540.6 (465.17)	541.48 (458.40)	540.00 (472.76)	0.019	0.985
Specialty Instrument				*χ*^2^_(2,145)_ = 5.902	0.052
Wind	49 (33.8)	24 (49.0)	25 (51.0)		
String	84 (57.9)	29 (34.5)	55 (65.5)		
Percussion	12 (8.3)	8 (66.7)	4 (33.3)		

Note: For quantitative variables, M (SD); for categorical variables, n (%).

**Table 2 behavsci-13-00722-t002:** Comparisons of perfectionism, self-efficacy, and resilience scores according to gender.

	Total	Men	Women	*t* _(_ * _df_ * _=143)_	*p*
	145	61 (42.4)	84 (57.9)		
Perfectionism	40.01 (10.62)	36.16 (9.33)	42.80 (10.68)	3.89	<0.001
Striving for Perfection	22.10 (5.70)	20.77 (5.34)	23.07 (5.79)	2.440	0.016
Negative Reactions	17.90 (6.59)	15.39 (5.59)	19.73 (6.69)	4.239	<0.001
Self-efficacy	31.29 (4.34)	30.84 (4.04)	31.62 (4.53)	1.074	0.284
Resilience (RS)	131.37 (18.00)	130.34 (17.15)	132.11 (18.67)	0.581	0.562
Personal competence	93.89 (12.55)	92.43 (12.35)	94.95 (12.66)	1.198	0.233
Acceptance of self and life	37.48 (7.24)	37.92 (6.59)	37.16 (7.71)	0.625	0.533
Categories in resilience				*χ*^2^_(1,145)_ = 0.345	0.557
High Resilience	27 (18.6)	10 (37.0)	17 (63.0)		
Low Resilience	118 (81.4)	51 (43.2)	67 (56.8)		

Note: For quantitative variables, M (SD); for categorical variables, n (%). Categories in resilience: low resilience (<147 RS points); high resilience (≥147 RS points).

**Table 3 behavsci-13-00722-t003:** Comparisons of perfectionism and self-efficacy according to the resilience categories.

		RS Categories			
	Total	High	Low	*t* _(_ * _df_ * _=143)_	*p*
	145	27 (18.6)	118 (81.4)		
Perfectionism	40.01 (10.62)	38.15 (9.86)	40.43 (10.78)	1.008	0.315
F1 Striving for Perfection	22.10 (5.70)	22.93 (4.91)	21.92 (5.87)	0.830	0.408
F2 Negative Reactions	17.90 (6.59)	15.22 (7.23)	18.52 (6.30)	2.382	0.019
Self-efficacy	31.29 (4.34)	35.22 (3.36)	30.39 (4.03)	5.785	<0.001

Note: For quantitative variables, M (SD); for categorical variables, n (%). Categories in resilience: low resilience (<147 RS points); high resilience (≥147 RS points).

**Table 4 behavsci-13-00722-t004:** Pearson’s bivariate correlations (resilience, perfectionism, self-efficacy).

	1	2	3	4	5	6	7
(1) RS-TOTAL	1						
(2) F1-RS	0.950/<0.001	1					
(3) F2-RS	0.840/<0.001	0.628/<0.001	1				
(4) PF TOTAL	−0.016/0.851	0.075/0.370	−0.169/0.042	1			
(5) F1 PF	0.136/0.102	0.207/0.013	−0.020/0.814	0.841/<0.001	1		
(6) F2 PF	−0.143/0.086	−0.058/0.489	−0.256/0.002	0.884/<0.001	0.491/<0.001	1	
(7) Self-Efficacy	0.525/<0.001	0.520/<0.001	0.403/<0.001	0.021/0.799	0.165/0.048	−0.108/0.195	1

Note: r/p; RS-TOTAL—total resilience score; F1-RS—Resilience Factor 1 (personal competence); F2-RS—Resilience Factor 2 (acceptance of self and life); PF TOTAL—Total perfectionism score; F1 PF—Perfectionism Factor 1 (striving for perfection); F2 PF—Perfectionism Factor 2 (negative reactions).

**Table 5 behavsci-13-00722-t005:** Stepwise regression analysis, taking resilience as the predicted variable and self-efficacy, functional perfectionism and dysfunctional perfectionism as predictor variables.

	*β*	*t*	*p*	*R* ^2^	Δ*R*^2^	*p*	*F*	*p*
Model 1				0.275	0.275	<0.001	*F*_(1,144)_ = 54.274	<0.001
Self-efficacy	0.525	7.367	<0.001					
Model 2				0.278	0.003	0.481	*F*_(2,144)_ = 27.292	<0.001
Self-efficacy	0.516	7.137	<0.001					
Striving for Perfection	0.051	0.706	0.481					
Model 3				0.295	0.018	0.063	*F*_(3,144)_ = 19.683	<0.001
Self-efficacy	0.486	6.612	<0.001					
Striving for Perfection	0.132	1.580	0.116					
Negative Reactions	−0.156	−1.872	0.063					

## Data Availability

The datasets generated during and/or analyzed during the current study are available from the corresponding author on reasonable request.

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
