# Peer review of "Resilient Behaviors in Music Students: Relationship with Perfectionism and Self-Efficacy"

_behavsci, 2023, doi:10.3390/bs13090722_

Round 1
Reviewer 1 Report
The research topic is relevant. The structure of the article meets the requirements for a scientific article. The theoretical overview of the main concepts is coherent and logically based. Research instruments and their measurement scales are described in the research methodology. The research results section raises the biggest concern regarding the correctness of the data analysis:
The scales of the research instruments were ordinal, but the Student’s criterion (t) was calculated, and Stepwise regression analysis was applied. In the case of regression analysis, the dependent (or predicted) variable must be in a ratio scale. Regression analysis is used to ascertain how independent variables cause dependent variables. Structural equation modeling (SEM) is recommended for hypothesis testing. I would like a more detailed explanation of what statistics and how they were used when testing the hypotheses.
Author Response
COMMENTS TO REVIEWERS
Title.- Resilient Behaviours in Music Students: Relationship with Perfectionism and Self-Efficacy
behavsci-2582953
BEHAVIORAL SCIENCES
--------------------------------------------------------------------------------------------------------------------
We would like to thank the reviewers for their comments and suggestions, which have undoubtedly improved the quality of the final version of this work.
We include the comments (in red type) to the suggestions made by the reviewers (in black type).
------------------------------------------------------------------------------------------
Reviewer #1:
The research topic is relevant. The structure of the article meets the requirements for a scientific article. The theoretical overview of the main concepts is coherent and logically based. Research instruments and their measurement scales are described in the research methodology. The research results section raises the biggest concern regarding the correctness of the data analysis:
1.- The scales of the research instruments were ordinal, but the Student’s criterion (t) was calculated, and Stepwise regression analysis was applied. In the case of regression analysis, the dependent (or predicted) variable must be in a ratio scale. Regression analysis is used to ascertain how independent variables cause dependent variables. Structural equation modeling (SEM) is recommended for hypothesis testing. I would like a more detailed explanation of what statistics and how they were used when testing the hypotheses.
RESPONSE.- Observational scale data allows for a wide range of statistical analyses. Once the necessary data is obtained, you can gather both descriptive and inferential statistics. Some of the most common parametric tests applied to test hypotheses involving ratio data include the t-test, ANOVA, Pearson's r for correlation, and simple linear regression.
Parametric tests are most appropriate when your ratio scale data follows a normal distribution. In our study, the data meets the criteria for normality (L-217-226). Using parametric tests enables you to draw stronger conclusions based on your data. Examples of common parametric tests include: 1) the use of the t-test, 2) ANOVA, 3) Pearson's r for correlation, and 4) simple linear regression for regression purposes.
Regression analysis is highly beneficial when guided by a theoretical model that suggests the potential causal order of variable influence and their entry sequence into the regression equation. However, its utility is limited when variables are correlated or when multiple variables are used within the same block, each exerting distinct facilitating or suppressing effects on the criterion variable (as seen in our study where adaptive and maladaptive perfectionism are correlated). To address these challenges, various methods can be employed, such as the stepwise variable selection technique within a predictive model (Peña, 2019). As Peña (2019) notes, practical experience suggests that good models generally align with reasonable criteria, and the ultimate model selection should be based on its logical alignment with the reality it describes.
Moreover, some authors suggest that operationalization could be feasible if scale items have at least 5 points on the measurement scale and are treated as interval or quasi-interval, with an underlying continuous construct (Bollen, 1989). Pell (2005) supports this view, asserting that groups of items can undergo parametric statistical analysis provided test conditions are met, leveraging their robust advantages. Furthermore, Carifio and Perla (2008) extend this idea, affirming the appropriateness of using parametric techniques with such variable types.
References:
Bhandari, P. (2023). Ratio Scales | Definition, Examples, & Data Analysis. Scribbr. Retrieved August 21, 2023, from https://www.scribbr.com/statistics/ratio-data/
Bollen, K.A. (1989). Structural Equations with Latent Variables. John Wiley and Sons.
Peña, D. (2019). Regresión y diseño de experimentos (1st ed., 4th reimp.). Madrid: Alianza Editorial, 2019.
Carifio, J., & Perla, R. (2008). Resolving the 50-year debate around using and misusing Likert scales. Medical education, 42(12), 1150–1152. https://doi.org/10.1111/j.1365-2923.2008.03172.x
Pell G. (2005). Use and misuse of Likert scales. Medical education, 39(9), 970–971. https://doi.org/10.1111/j.1365-2929.2005.02237.x
------------------------------------------------------------------------------------------
We hope that the changes made according to the suggestions of the Editor and Reviewers are satisfactory.
Kind regards,
Reviewer 2 Report
Dear Author,
I am pleased to review an original paper draft entitled "Resilient Behaviours in Music Students: Relationship with Perfectionism and Self-Efficacy", it is an interesting quantitative study built at an intersection of Multidimensional Inventory of Perfectionism in Sport, resilience using the Resilience Scale, and self-efficacy using the General Self-Efficacy Scale. In general, this manuscript is attractive, scientifically sound, and very promising. It has a number of theoretical and practical implications. However, there is room for improvement, please follow my points below:
1. (17-18) Findings, results, and conclusions should be disclosed and briefly explained in the abstract. Now it looks too general and uninformative.
2. Compelling introduction! Very well-structured and comprehensive, with great flow! However, (30-43) I would suggest to involve the self-concept before start articulating self-efficacy. Namely, please see:
Glebova, E.; López-Carril, S. ‘Zero Gravity’: Impact of COVID-19 Pandemic on the Professional Intentions and Career Pathway Vision of Sport Management Students. Educ. Sci. 2023, 13, 807. https://doi.org/10.3390/educsci13080807
Asmus, E.P., 2021. Motivation in music teaching and learning. Visions of Research in Music Education, 16(5), p.31.
3. (95-106) This paragraph fits perfectly, but I would suggest explaining hypotheses in greater detail, if possible.
4. (108-115) Could you please justify your sample? Why this category of people has been chosen for the test?
5. (155-161) I would be interested to know about the data validation procedure, if any... please indicate
6. Discussion is well articulated and informative, however, subsections could help readers' navigation.
7. Any limitations? Future/ recommended research directions?
Thank you for your attention, I hope, you find these comments helpful.
Author Response
COMMENTS TO REVIEWERS
Title.- Resilient Behaviours in Music Students: Relationship with Perfectionism and Self-Efficacy
behavsci-2582953
BEHAVIORAL SCIENCES
--------------------------------------------------------------------------------------------------------------------
We would like to thank the reviewers for their comments and suggestions, which have undoubtedly improved the quality of the final version of this work.
We include the comments (in red type) to the suggestions made by the reviewers (in black type).
------------------------------------------------------------------------------------------
Reviewer #2:
I am pleased to review an original paper draft entitled "Resilient Behaviours in Music Students: Relationship with Perfectionism and Self-Efficacy", it is an interesting quantitative study built at an intersection of Multidimensional Inventory of Perfectionism in Sport, resilience using the Resilience Scale, and self-efficacy using the General Self-Efficacy Scale. In general, this manuscript is attractive, scientifically sound, and very promising. It has a number of theoretical and practical implications. However, there is room for improvement, please follow my points below:
- (17-18) Findings, results, and conclusions should be disclosed and briefly explained in the abstract. Now it looks too general and uninformative.
RESPONSE.- Due to word limitations in the summary, the results have been made explicit in a general way. However, we have expanded the abstract and made the results more specific as suggested by the reviewer (L-12-19).
- Compelling introduction! Very well-structured and comprehensive, with great flow! However, (30-43) I would suggest to involve the self-concept before start articulating self-efficacy. Namely, please see:
Glebova, E.; López-Carril, S. ‘Zero Gravity’: Impact of COVID-19 Pandemic on the Professional Intentions and Career Pathway Vision of Sport Management Students. Educ. Sci. 2023, 13, 807. https://doi.org/10.3390/educsci13080807
Asmus, E.P., 2021. Motivation in music teaching and learning. Visions of Research in Music Education, 16(5), p.31.
RESPONSE.- When we speak of self-concept we refer to the characteristics that define us; there is not necessarily an evaluation but rather a description. On the other hand, when we use the term self-efficacy, we are referring to the areas in which we think we are competent (or not) or that we are good (or bad) at; we are referring to our capabilities. To introduce self-concept would be to introduce one more variable that will not be analyzed. In this work we have not attempted to analyze how we define ourselves, which is why self-concept has not been considered. However, self-efficacy has been widely studied in the context of high performance.
The reviewer's comment is noted and has been included as a suggestion for future work (L.- 349-358)
- (95-106) This paragraph fits perfectly, but I would suggest explaining hypotheses in greater detail, if possible.
RESPONSE.- Thanks for the comment. Some lines have been introduced that seek to support what has been said in the hypotheses
- (108-115) Could you please justify your sample? Why this category of people has been chosen for the test?
RESPONSE.- Thanks for the suggestion. You have specified (L-104-110) some of the reasons that led to select this performance group.
- (155-161) I would be interested to know about the data validation procedure, if any... please indicate
RESPONSE.- We greatly appreciate your interest in the data validation procedures conducted for this study. We have now incorporated a dedicated section in the article that provides comprehensive details regarding the implemented data validation processes (178-185). This section outlines the steps taken to ensure the quality and reliability of the data used in our analysis, including verification, cleaning, imputation techniques for addressing missing values, and exploratory data analysis to identify any anomalies. By including this information, we aim to enhance the transparency and rigor of our study's methodology, reassuring readers about the integrity of our findings. We hope you find this addition valuable and thank you for your insightful feedback."
- Discussion is well articulated and informative, however, subsections could help readers' navigation.
RESPONSE.- Thanks for the suggestion. Effectively, generating sections is an option that could be helpful. However, we have decided to keep a single section since it maintains a coherent argument in the development of the Discussion.
- Any limitations? Future/ recommended research directions?
RESPONSE.- Thank you for your comment. A paragraph has been inserted (L.- 349-358)
Thank you for your attention, I hope, you find these comments helpful.
------------------------------------------------------------------------------------------
We hope that the changes made according to the suggestions of the Editor and Reviewers are satisfactory.
Kind regards,
Round 2
Reviewer 2 Report
Dear Author,
Thank you for the effective revisions. To complete the round of revisions, I would suggest accompanying the future research directions with literature references.
Author Response
COMMENTS TO REVIEWER-2-R2
Title.- Resilient Behaviours in Music Students: Relationship with Perfectionism and Self-Efficacy
behavsci-2582953
BEHAVIORAL SCIENCES
--------------------------------------------------------------------------------------------------------------------
We would like to thank the reviewers for their comments and suggestions, which have undoubtedly improved the quality of the final version of this work.
We include the comments (in red type) to the suggestions made by the reviewers (in black type).
------------------------------------------------------------------------------------------
Reviewer #2: R2
Thank you for the effective revisions. To complete the round of revisions, I would suggest accompanying the future research directions with literature references.
RESPONSE.- For future research, it is necessary to establish experimental designs to define the causal relationships between the type of perfectionism (adaptive/disadaptive) and resilient behaviors; as some researchers have argued that higher performance may be a factor contributing to the development of perfectionism in children and adolescents [59]. In addition, it is necessary to understand how maladaptive perfectionism interferes with or hinders performance (in the short and long term) and adjustment to situations with ad-verse outcomes. In this regard, it is important to note the number of research studies that support the benefits of perfectionism, provided that perfectionistic efforts are not accompanied by high levels of perfectionistic concerns. It has been suggested that perfectionistic concerns are unhealthy and maladaptive and can pose a serious risk to people's well-being and mental health.
On the other hand, it is necessary to control for factors such as the influence of the teacher (other-oriented perfectionism) [11], the type of performance of the participants or the type of instruments used, since differences have been shown depending on the instrument considered [60]. Future research could examine whether the interaction between types of perfectionism and coping strategies differs as a result of the musician's experience [61].
59 Flett, G.L.; Hewitt, P.L.; Oliver, J.M.; Macdonald, S. Perfectionism in children and their parents: A developmental analysis. In G. L. Flett & P. L. Hewitt (Eds.), Perfectionism: Theory, research, and treatment (pp. 89-132). Washington, DC: American Psychological Association. 2002.
60 Kenny, D.T. The psychology of music performance anxiety. Oxford University Press. 2011.
61 McNeil, D.; Loi, N.; Bullen, R. Investigating the moderating role of coping style on music performance anxiety and perfectionism. International Journal of Music Education, 2022,40(4): 587-597. https://doi.org/10.1177/02557614221080523
------------------------------------------------------------------------------------------
We hope that the changes made according to the suggestions of the Editor and Reviewers are satisfactory.
Kind regards,